# Nonparametric Uncertainty Quantification for Single Deterministic Neural Network

**Nikita Kotelevskii**[1], **Aleksandr Artemenkov**[1], **Kirill Fedyanin**[2],
**Fedor Noskov**[1,4], **Alexander Fishkov**[1], **Artem Shelmanov**[3,7],
**Artem Vazhentsev**[1,3], **Aleksandr Petiushko**[5,6], and **Maxim Panov**[2]

[1]Skolkovo Institute of Science and Technology, Moscow, Russia
[2]Technology Innovation Institute, Abu Dhabi, UAE
[3]AIRI, Moscow, Russia
[4]HSE University, Moscow, Russia
[5]Lomonosov Moscow State University, Moscow, Russia
[6]Nuro, Inc.
[7]Mohamed bin Zayed University of Artificial Intelligence, Abu Dhabi, UAE

## Abstract

This paper proposes a fast and scalable method for uncertainty quantification of machine learning models' predictions. First, we show the principled way to measure the uncertainty of predictions for a classifier based on Nadaraya-Watson's nonparametric estimate of the conditional label distribution. Importantly, the proposed approach allows to disentangle explicitly *aleatoric* and *epistemic* uncertainties. The resulting method works directly in the feature space. However, one can apply it to any neural network by considering an embedding of the data induced by the network. We demonstrate the strong performance of the method in uncertainty estimation tasks on text classification problems and a variety of real-world image datasets, such as MNIST, SVHN, CIFAR-100 and several versions of ImageNet.

## 1 Introduction

In many machine learning applications, it is crucial to complement model predictions with uncertainty scores that reflect the degree of trust in these predictions. The total uncertainty of a prediction sums from two uncertainty types, arising from different sources: *aleatoric* and *epistemic* [14, 33]. The former reflects the irreducible noise and ambiguity in the data due to class overlap, while the latter is related to the lack of knowledge about model parameters, and can be reduced by expanding the training dataset. Disentangling epistemic uncertainty can help to identify *out-of-distribution (OOD) data* or to spot instances important for annotation during active learning [22]. The areas with high aleatoric uncertainty might contain some incorrectly labeled instances. If we quantify both types of uncertainty well, we can effectively *abstain from predictions* in unreliable areas and address a decision to a human expert [18], which is important in safe-critical fields, such as medicine [53], autonomous driving [43, 20], and finance [6].

There is no universally accepted uncertainty measure, and diverse, often heuristic treatments are used in practice. For this purpose, one simply could use maximum softmax probabilities of deep neural network (NN). However, MaxProb represents only aleatoric uncertainty, and the resulting measure is notorious to be overconfident in data areas the model did not see during training [59]. Methods based on ensembling [37] or Bayesian techniques [21] can capture both types of uncertainty and yield more

---

Correspondence to: <`Nikita.Kotelevskii@skoltech.ru`> and <`Maxim.Panov@tii.ae`>

36th Conference on Neural Information Processing Systems (NeurIPS 2022).

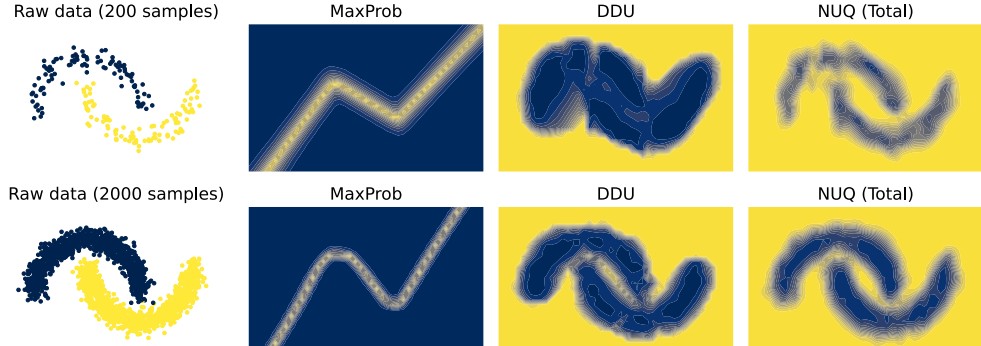

Figure 1: The first column on the left shows the raw data from the Two Moons dataset. The second column illustrates neural network prediction uncertainty obtained using MaxProb. The last two columns illustrate the results of DDU and the sum of aleatoric and epistemic uncertainties obtained using NUQ. The lighter color indicates higher uncertainty.

reliable uncertainty estimates (UEs)[1]. However, along with that, they introduce large computational overhead and might require big modifications to a model architecture and a training procedure.

Recently, a series of UE methods based on a single deterministic neural network model has been developed [41, 69, 46, 70]. The main idea behind these methods is to leverage geometrical proximity of instances in a vector space of neural network hidden representations that capture semantic relationships between instances in the input space. Such methods are very promising since they are computationally efficient and usually do not require big changes in network architectures and training procedures, which gives them versatility.

Many of these methods rely on the assumption that (conditionally on the class) training instances are distributed normally in the latent space of a trained neural network [70, 41]. However, this assumption does not always hold. For example, a border between classes might be more complex, as well as the border between the in-domain and the out-of-domain region. This happens when training is accomplished in a low-resource regime with small amounts of labeled instances. Another problem is that some methods are hard to train and the model performance can substantially deteriorate without proper hyperparameter tuning. Finally, some methods require computing the covariance matrix of training data [41]. This operation might be computationally unstable if the training dataset is not large enough.

In this work, we propose a new principled approach to UE that overcomes these limitations and demonstrates very robust performance both in low-resource and in high-resource regimes. We suggest to look at the pointwise Bayes risk (probability of the wrong prediction) as the natural measure of the model prediction uncertainty at a particular data point. Then we consider the Nadaraya-Watson estimator of the conditional label distribution. Its asymptotic Gaussian approximation allows deriving uncertainty estimate based on the upper bound for the risk. The resulting Nonparametric Uncertainty Quantification (NUQ) method is amenable for uncertainty disentanglement (into aleatoric and epistemic) and is implemented in a scalable manner, which allows it to be used on large datasets such as ImageNet.

To illustrate the strong points of the proposed method, we suggest to look at a toy example in Figure 1. Here, we compare NUQ with MaxProb and DDU [55], one of the strong baselines in deterministic uncertainty estimation aimed at epistemic uncertainty. MaxProb as a measure of aleatoric uncertainty gives high values only in-between classes while being very confident far from the data. We also see that for a larger data set of 2000 points NUQ and DDU show similar results. However, when the neural network model is trained only on 200 labeled instances, DDU scores become very rough. It cannot spot an unreliable area in-between classes, while the NUQ-based total uncertainty measure does it precisely with high uncertainty score. Importantly, the presented total uncertainty for NUQ can be also disentangled into aleatoric and epistemic ones.

---

[1]Terms uncertainty quantification and uncertainty estimation are often used in literature interchangeably.

Below, we give a **summary of contributions** of this paper.

1. We develop *a new and theoretically grounded* method for nonparametric uncertainty quantification (see Section 2), which is (a) applicable to any deterministic neural network model, (b) has an efficient implementation, (c) allows to disentangle aleatoric and epistemic uncertainty, (d) outperforms other techniques on image and text classification tasks (see Section 6).

2. We formally prove the consistency of the NUQ-based estimator when applied to the problem of classification with the reject option, see Section 4.

3. We conduct a vast empirical investigation of NUQ and other UE techniques on image and text classification tasks that supports our claims, see Section 6.

## 2 Nonparametric Uncertainty Quantification

### 2.1 Classification under Covariate Shift

In this section and below, for the sake of clarity, we provide derivations for the binary case. We derive a generalization to a multi-class setting in Supplementary Material (SM), Section A.

Let us consider the standard binary classification setup $(X, Y) \in \mathbb{R}^d \times \{0, 1\}$ with $(X, Y) \sim \mathbb{P}$, where $\mathbb{P}$ is a joint distribution of objects and the corresponding labels. We assume that we observe the dataset $\mathcal{D} = \left\{ (\mathbf{x}_i, y_i) \right\}_{i=1}^N$ of *i.i.d.* points from $\mathbb{P} = \mathbb{P}_{\text{train}}$. Here, $\mathbf{x}$ and $y$ are realizations of random variables $X$ and $Y$.

The classical problem in statistics and machine learning is using the dataset $\mathcal{D}$, to find a rule $\hat{g}$, which approximates the optimal one:

$$g^* = \arg \min_g \mathbb{P}(g(X) \neq Y).$$

Here, $g \colon \mathbb{R}^d \to \{0, 1\}$ is any classifier, and the probability of wrong classification $\mathbb{P}(g(X) \neq Y)$ is usually called *risk*. The rule $g^*$ is given by the *Bayes optimal classifier*:

$$g^*(\mathbf{x}) = \begin{cases} 1, & \eta(\mathbf{x}) \geq \frac{1}{2}, \\ 0, & \eta(\mathbf{x}) < \frac{1}{2}, \end{cases}$$

where $\eta(\mathbf{x}) = \mathbb{P}(Y = 1 \mid X = \mathbf{x})$ which is the conditional probability of $Y$ given $X = \mathbf{x}$ under the distribution $\mathbb{P}$.

In this work, we consider a situation when the distribution of the test samples $\mathbb{P}_{\text{test}}$ is different from the one for the training dataset $\mathbb{P}_{\text{train}}$, i.e. $\mathbb{P}_{\text{test}} \neq \mathbb{P}_{\text{train}}$. Obviously, the rule $g^*$ obtained for $\mathbb{P} = \mathbb{P}_{\text{train}}$ might no longer be optimal if the aim is to minimize the error on the test data $\mathbb{P}_{\text{test}}(g(X) \neq Y)$.

In order to formulate a meaningful estimation problem, some additional assumptions are needed. We assume that the conditional label distribution $\eta(\mathbf{x}) = \mathbb{P}(Y \mid X = \mathbf{x})$ **is the same** under both $\mathbb{P}_{\text{train}}$ and $\mathbb{P}_{\text{test}}$. This assumption has two important consequences:

1. The entire difference between $\mathbb{P}_{\text{train}}$ and $\mathbb{P}_{\text{test}}$ is due to the difference between marginal distributions of $X$: $p_{\text{train}}(X)$ and $p_{\text{test}}(X)$. The situation when $p_{\text{test}}(X) \neq p_{\text{train}}(X)$ is known as *covariate shift*.

2. The rule $g^*$ is still valid, i.e., optimal under $\mathbb{P}_{\text{test}}$.

However, while the classifier $g^*$ is still optimal under covariate shift, its approximation $\hat{g}$ might be arbitrarily bad. The reason for that is that we cannot expect $\hat{g}$ to approximate $g^*$ well in the areas where we have few objects from the training set or do not have them at all. Thus, some special treatment of the covariate shift is required.

### 2.2 Pointwise Risk and Its Estimation

We consider a classification rule $\hat{g}(X) = \hat{g}_\mathcal{D}(X)$ constructed based on the dataset $\mathcal{D}$. Let us start from defining the pointwise risk of a prediction:

$$\mathcal{R}(\mathbf{x}) = \mathbb{P}(\hat{g}(X) \neq Y \mid X = \mathbf{x}),$$

where $\mathbb{P}(\hat{g}(X) \neq Y \mid X = \mathbf{x}) \equiv \mathbb{P}_{\text{train}}(\hat{g}(X) \neq Y \mid X = \mathbf{x}) \equiv \mathbb{P}_{\text{test}}(\hat{g}(X) \neq Y \mid X = \mathbf{x})$ under the assumptions above. The value $\mathcal{R}(\mathbf{x})$ is independent of the covariate distribution $p_{\text{test}}(X)$ and essentially allows to define a meaningful target of estimation, which is based solely on the quantities known for the training distribution.

Let us note that the total risk value $\mathcal{R}(\mathbf{x})$ admits the following decomposition:

$$\mathcal{R}(\mathbf{x}) = \tilde{\mathcal{R}}(\mathbf{x}) + \mathcal{R}^*(\mathbf{x}),$$

where $\mathcal{R}^*(\mathbf{x}) = \mathbb{P}(g^*(X) \neq Y \mid X = \mathbf{x})$ is the Bayes risk and $\tilde{\mathcal{R}}(\mathbf{x}) = \mathbb{P}(\hat{g}(X) \neq Y \mid X = \mathbf{x}) - \mathbb{P}(g^*(X) \neq Y \mid X = \mathbf{x})$ is an excess risk. Here, $\mathcal{R}^*(\mathbf{x})$ corresponds to aleatoric uncertainty as it completely depends on the data distribution. The excess risk $\tilde{\mathcal{R}}(\mathbf{x})$ directly measures imperfectness of the model $\hat{g}$ and, thus, can be seen as a measure of epistemic uncertainty.

To proceed, we first assume that the classifier $\hat{g}$ has the standard form:

$$\hat{g}(\mathbf{x}) = \begin{cases} 1, & \hat{\eta}(\mathbf{x}) \geq \frac{1}{2}, \\ 0, & \hat{\eta}(\mathbf{x}) < \frac{1}{2}, \end{cases}$$

where $\hat{\eta}(\mathbf{x}) = \hat{p}(Y = 1 \mid X = \mathbf{x})$ is an estimate of the conditional density $\eta(\mathbf{x})$.

For such an estimate, we can upper bound the excess risk via the following classical inequality [15]:

$$\tilde{\mathcal{R}}(\mathbf{x}) = \mathbb{P}(\hat{g}(X) \neq Y \mid X = \mathbf{x}) - \mathbb{P}(g^*(X) \neq Y \mid X = \mathbf{x}) \leq 2|\hat{\eta}(\mathbf{x}) - \eta(\mathbf{x})|.$$

It allows us to obtain an upper bound for the total risk:

$$\mathcal{R}(\mathbf{x}) \leq \mathcal{L}(\mathbf{x}) = \mathcal{R}^*(\mathbf{x}) + 2|\hat{\eta}(\mathbf{x}) - \eta(\mathbf{x})|,$$

where $\mathcal{R}^*(\mathbf{x}) = \min\{\eta(\mathbf{x}), 1 - \eta(\mathbf{x})\}$ in the case of binary classification. While this upper bound still depends on the unknown quantity $\eta(\mathbf{x})$, we will see in the next section that $\mathcal{L}(\mathbf{x})$ allows for an efficient approximation under mild assumptions.

### 2.3 Nonparametric Uncertainty Quantification

#### 2.3.1 Kernel Density Estimate and Its Asymptotic Distribution

To obtain an estimate of $\mathcal{L}(\mathbf{x})$ and, consequently, bound the risk, we need to consider some particular type of estimator $\hat{\eta}$. In this work, we choose the classical kernel-based Nadaraya-Watson estimator of the conditional label distribution as it allows for a simple description of its asymptotic properties.

Let us denote by $K_h \colon \mathbb{R}^d \mapsto \mathbb{R}$ the multi-dimensional kernel function with bandwidth $h$. Typically, we consider a multi-dimensional Gaussian kernel, but other choices are also possible.

The conditional probability estimate is expressed as ($y_i$ is either 0 or 1):

$$\hat{\eta}(\mathbf{x}) = \frac{\sum_{i=1}^{N} \mathbb{1}[y_i = 1] \cdot K_h(\mathbf{x} - \mathbf{x}_i)}{\sum_{i=1}^{N} K_h(\mathbf{x} - \mathbf{x}_i)}. \tag{1}$$

The difference between $\hat{\eta}(\mathbf{x}) - \eta(\mathbf{x})$ for properly chosen bandwidth $h$ converges in distribution as follows (see, e.g. [63]):

$$\hat{\eta}(\mathbf{x}) - \eta(\mathbf{x}) \to \mathcal{N}\left(0, \frac{\tilde{C}}{N}\frac{\sigma^2(\mathbf{x})}{p(\mathbf{x})}\right), \tag{2}$$

where $N$ is the number of data points in the training set, $p(\mathbf{x})$ is the marginal distribution of covariates (see details in SM, Section C.3), and $\sigma^2(\mathbf{x})$ is the standard deviation of the data label at point $\mathbf{x}$. For binary classification, $\sigma^2(\mathbf{x}) = \eta(\mathbf{x})(1 - \eta(\mathbf{x}))$. The constant $\tilde{C} = \int [K_h(\mathbf{u})]^2 d\mathbf{u}$, where $\mathbf{u}$ is an integration variable, depends only on the choice of the kernel $K_h$ and could be computed in closed form for popular kernels. See in details in SM, Section C.4.

Now, we are equipped with an estimate of the distribution for $\hat{\eta}(\mathbf{x}) - \eta(\mathbf{x})$. Let us denote by $\tau(\mathbf{x})$ the standard deviation of a Gaussian from the equation (2):

$$\tau^2(\mathbf{x}) = \frac{\tilde{C}}{N}\frac{\sigma^2(\mathbf{x})}{p(\mathbf{x})}.$$

In the following sections, we first show how to use the obtained property for uncertainty estimation, and then, we show how it can be computed.

### 2.3.2 Total, Aleatoric, and Epistemic Uncertainty and Their Estimates

In this work, we suggest a particular uncertainty quantification procedure inspired by the derivation above, which we call *Nonparametric Uncertainty Quantification (NUQ)*. More specifically, we suggest to consider the following measure of the total uncertainty:

$$\mathbf{U}_t(\mathbf{x}) = \min\{\eta(\mathbf{x}), 1 - \eta(\mathbf{x})\} + 2\sqrt{\frac{2}{\pi}}\tau(\mathbf{x}).$$

This measure is obtained by considering an asymptotic approximation of the expected value of the total risk upper bound:

$$\mathbb{E}_{\mathcal{D}}\mathcal{L}(\mathbf{x}) = \min\{\eta(\mathbf{x}), 1 - \eta(\mathbf{x})\} + 2\mathbb{E}_{\mathcal{D}}|\hat{\eta}(\mathbf{x}) - \eta(\mathbf{x})|$$

in view of (2) and the fact that $\mathbb{E}|\xi| = \text{std}(\xi)\sqrt{\frac{2}{\pi}}$ for the zero-mean normal variable $\xi$. The resulting estimate upper bounds the average error of estimation at the point $\mathbf{x}$ and thus indeed can be used as the measure of total uncertainty.

We also can write the corresponding measures of aleatoric and epistemic uncertainties:

$$\mathbf{U}_a(\mathbf{x}) = \min\{\eta(\mathbf{x}), 1 - \eta(\mathbf{x})\}, \quad \mathbf{U}_e(\mathbf{x}) = 2\sqrt{\frac{2}{\pi}}\tau(\mathbf{x}).$$

Finally, the data-driven uncertainty estimates $\hat{\mathbf{U}}_a(\mathbf{x}), \hat{\mathbf{U}}_e(\mathbf{x})$ and $\hat{\mathbf{U}}_t(\mathbf{x})$ can be obtained via plug-in using estimates $\hat{\eta}(\mathbf{x}), \hat{\sigma}(\mathbf{x}), \hat{p}(\mathbf{x})$ and, consequently, $\hat{\tau}^2(\mathbf{x}) = \frac{1}{N}\frac{\hat{\sigma}^2(\mathbf{x})}{\hat{p}(\mathbf{x})}\tilde{C}$.

We should note that despite being based on asymptotic approximation, the resulting formulas for uncertainties are very natural and make sense for finite sample size (for example, epistemic uncertainty is proportional to $\sigma^2(\mathbf{x})/p(\mathbf{x})$). We also should note that even known non-asymptotic decompositions for the risk of the NW-estimator still contain the same $\sigma^2(\mathbf{x})/p(\mathbf{x})$ component, see Proposition 1 in [7]. That means that the proposed estimate well captures the general uncertainty trend.

**Efficient computation.** We note that computation of the nonparametric estimate (1) involves a sum over the whole available data. This could be intractable in practice when we are working with large datasets. However, the typical kernel $K_h$ quickly approaches zero with the increase of the norm of the argument: $\|\mathbf{x} - \mathbf{x}_i\|$. Thus, we can use an approximation of the kernel estimate: instead of the sum over all elements in the dataset, we consider the contribution of only several nearest neighbors (see SM, Section C.2 for details). It requires a fast algorithm for finding the nearest neighbors. For this purpose, we use the approach of [51] based on Hierarchical Navigable Small World graphs (HNSW). It provides a fast, scalable, and easy-to-use solution to the computation of nearest neighbors.

**Application to NN and Comparison with Existing Methods.** The resulting NUQ method can be applied to NN in the postprocessing fashion, i.e. one can fit it on top of the embeddings of the trained NN model. One may wonder about the difference between NUQ and other embedding based methods such as, for example, DUQ [69] or DDU [55]. The difference is twofold: (i) NUQ is based on the rigorous derivation of total, aleatoric, and epistemic uncertainties, while other methods usually consider more heuristic treatment and do not allow for uncertainty disentanglement; (ii) NUQ considers a more flexible estimator of density in the embedding space that allows to achieve better quality in a small training data regime or for complicated data; see experimental evaluation in Section 6.

## 3 Detailed Algorithmic Description of NUQ Approach

In this section, we provide a detailed algorithmic description of the NUQ approach. On the training stage, NUQ uses training data embeddings obtained from the pre-trained neural network and builds a Bayesian classifier based on conditional label probabilities estimated in a non-parametric way:

$$\hat{p}(Y = c \mid X = \mathbf{x}) = \frac{\sum_{i=1}^{N} \mathbb{1}[y_i = c] \cdot K_h(\mathbf{x} - \mathbf{x}_i)}{\sum_{i=1}^{N} K_h(\mathbf{x} - \mathbf{x}_i)}, \quad c = 1, \ldots, C.$$

The bandwidth $h$ is tuned via cross-validation optimizing the classification accuracy on the training data (see SM, Section C.1).

**Algorithm 1** NUQ inference algorithm.

---

**Input:** Training set $\{(\mathbf{x}_i, y_i)\}_{i=1}^N$, inference point $\mathbf{x}$, bandwidth $h$
**Output:** Prediction $\hat{g}(\mathbf{x})$ and uncertainty estimate $\hat{\mathbf{U}}_t(\mathbf{x})$

$\{\mathbf{x}_{i_k}\}_{k=1}^K \leftarrow K$ nearest neighbors of $\mathbf{x}$ among $\{\mathbf{x}_i\}_{i=1}^N$

$\hat{p}(Y = c \mid X = \mathbf{x}) \leftarrow \frac{\sum_{k=1}^K K_h(\mathbf{x}_{i_k} - \mathbf{x})\mathbb{1}_{[y_{i_k} = c]}}{\sum_{k=1}^K K_h(\mathbf{x}_{i_k} - \mathbf{x})}$

$\hat{\sigma}_c^2(\mathbf{x}) = \hat{p}(Y = c \mid X = \mathbf{x})\big(1 - \hat{p}(Y = c \mid X = \mathbf{x})\big)$

$\hat{g}(\mathbf{x}) \leftarrow \underset{c}{\arg\max}\ \hat{p}(Y = c \mid X = \mathbf{x})$

$\hat{p}(\mathbf{x}) \leftarrow$ either KDE: $\frac{1}{Nh^d} \sum_{k=1}^K K_h(\mathbf{x}_{i_k} - \mathbf{x})$ or GMM

$\hat{\tau}^2(\mathbf{x}) \leftarrow \frac{1}{N} \frac{\max_c \hat{\sigma}_c^2(\mathbf{x})}{\hat{p}(\mathbf{x})} \tilde{C}$, where $\tilde{C} = \int [K_h(\mathbf{u})]^2 d\mathbf{u}$

$\hat{\mathbf{U}}_t(\mathbf{x}) \leftarrow \min_c\big\{1 - \hat{p}(Y = c \mid X = \mathbf{x})\big\} + 2\sqrt{\frac{2}{\pi}}\hat{\tau}(\mathbf{x})$

---

On the inference stage, the new object is passed through the neural network and a corresponding embedding is computed. Then, this embedding is used to compute the uncertainty estimates employing the estimated bandwidth $h$, see Algorithm 1 that details the computation of all the necessary intermediate quantities as well as the resulting uncertainty estimates based on the embeddings $\mathbf{x}$ provided by the neural network. Algorithm 1 also takes into account the usage of nearest neighbours to speed up the computation of kernel-based estimates. Finally, the computation of an estimate of the embeddings density $\hat{p}(\mathbf{x})$ can be done either via the kernel density estimate (KDE) or via the Gaussian Mixture Model (GMM). We study the relative benefits of these approaches in Section F.4.

## 4 Consistency of NUQ-based Classification with a Reject Option

Above, we obtained uncertainty estimates that characterize the classical risk of a prediction. However, they are also helpful to solve the formal problem of classification with the reject option. In this problem, for any input $\mathbf{x}$ we can choose either we perform prediction or reject it. Following [8], we assume that in the case of prediction we pay a binary price depending whether the prediction was correct or not, while in the case of the rejection we pay the constant price $\lambda \in (0, 1)$. For this task, the risk function is

$$\mathcal{R}_\lambda(\mathbf{x}) = \mathcal{R}(\mathbf{x})\mathbb{1}\{\alpha(\mathbf{x}) = 0\} + \lambda\mathbb{1}\{\alpha(\mathbf{x}) = 1\},$$

where $\alpha(\mathbf{x})$ is an indicator of the rejection.

The minimizer of $\mathcal{R}_\lambda(\mathbf{x})$ is given by the optimal Bayes classifier $g^*(\mathbf{x})$ and the abstention function

$$\alpha^*(\mathbf{x}) = \begin{cases} 0, & \mathcal{R}^*(\mathbf{x}) \le \lambda, \\ 1, & \mathcal{R}^*(\mathbf{x}) > \lambda. \end{cases}$$

To approximate $\alpha^*(\mathbf{x})$, we utilize hypothesis testing:

$$H_0 \colon \mathcal{R}^*(\mathbf{x}) > \lambda \text{ vs. } H_1 \colon \mathcal{R}^*(\mathbf{x}) \le \lambda.$$

We choose the confidence level $\beta > 0$ and consider the statistic $\hat{\mathbf{U}}_\beta(\mathbf{x}) = \min\{\hat{\eta}(\mathbf{x}), 1 - \hat{\eta}(\mathbf{x})\} + z_{1-\beta}\hat{\tau}(\mathbf{x})$, where $z_{1-\beta}$ is the $1 - \beta$ quantile of the standard normal distribution. The statistic $\hat{\mathbf{U}}_\beta(\mathbf{x})$ combines the plug-in estimate of the Bayes risk $\min\{\hat{\eta}(\mathbf{x}), 1 - \hat{\eta}(\mathbf{x})\}$ and the term $z_{1-\beta}\hat{\tau}(\mathbf{x})$ accounting for the confidence of estimation. The resulting abstention rule is given by:

$$\hat{\alpha}_\beta(\mathbf{x}) = \begin{cases} 0, & \hat{\mathbf{U}}_\beta(\mathbf{x}) \le \lambda, \\ 1, & \hat{\mathbf{U}}_\beta(\mathbf{x}) > \lambda. \end{cases}$$

Finally, if we consider the pair of kernel classifier $\hat{g}(\mathbf{x})$ and $\hat{\alpha}_\beta(\mathbf{x})$ then, we can prove the consistency result for the corresponding risk $\hat{\mathcal{R}}_\lambda(\mathbf{x})$ under standard assumptions on nonparametric densities, see SM, Section D for details.

**Theorem 4.1.** *Suppose that assumptions D.1–D.3 hold and $p(\mathbf{x}) > 0$, the bandwidth $h \to 0$ and $Nh^d \to \infty$ as $N$ tends to infinity. Then, for any $\beta < 1/2$:*

$$\mathbb{E}_\mathcal{D}\hat{\mathcal{R}}_\lambda(\mathbf{x}) - \mathcal{R}_\lambda^*(\mathbf{x}) \underset{N\to\infty}{\longrightarrow} 0.$$

This result shows the validity of the NUQ-based abstention procedure. Interesting future work is to obtain a precise convergence rate for the method. It should be possible based on the finite sample bounds provided in SM, Section D. We also experimentally illustrate the benefits of the proposed estimator in SM, Section D.1.

## 5 Related Work

The notion of uncertainty naturally appears in Bayesian statistics [25], and, thus, Bayesian methods are often used for uncertainty quantification. The exact Bayesian inference is computationally intractable, and approximations are used. Two popular ideas are the Markov Chain Monte Carlo sampling (MCMC; [57]) and the Variational Inference (VI; [4]). MCMC has theoretical guarantees to be asymptotically unbiased, but it has a high computational cost. VI-based approaches [64, 16, 60, 35] are more scalable, they are biased and at least double the number of parameters. That is why some alternatives are considered, such as the Bayesian treatment of Monte-Carlo dropout [21].

Deep Ensemble [37] is usually considered as a quite strong yet expensive approach. A series of papers developed ways of approximating the distribution obtained using an ensemble of models by a single probabilistic model [49, 50, 66]. These methods require changing the training procedure and need more parameters to train.

Recently, a series of uncertainty quantification approaches for a single deterministic neural network model was proposed. In DUQ [69], an RBF layer is added to the network with a custom training procedure to adjust the centroid points (in the embedding space). The downside of the method is its inability to distinguish aleatoric and epistemic uncertainty. Another approach to capture epistemic uncertainty was proposed in DDU [55]. It uses a Gaussian mixture model to estimate the density of objects in the embedding space of a trained neural network. The density values are then used as a confidence measure. SNGP [46] and DUE [70] are similar but use a Gaussian process as the final layer, requiring estimating covariance with the use of inducing points or RFF expansion.

There is a wide range of papers discussing classification with the reject option. Most likely, the problem was firstly studied by Chow in [9, 8]. Moreover, in [8], he introduced a risk function used across this paper. Herbei et al. [31] studied an optimal procedure for this risk and provided a plug-in rule. In the following works, empirical risk minimization among a class of hypotheses (see [2, 10]) or other types of risk (see [13, 19, 42]) were investigated. Besides, a number of practical works have been presented, see, for example, [27, 24, 56].

## 6 Experiments

We conduct a series of experiments on image and text classification datasets. In each experiment, (1) we train a parametric model – a neural network, which we call a *base model*; (2) fit NUQ on the training data using the embeddings obtained from the base model. We use logits as extracted features, if not explicitly stated otherwise. However, other options are also possible; see Section F.4.

Following SNGP and DDU, we use spectral normalization to train the base model to achieve bi-Lipschitz property and avoid the feature collapse and non-smoothness. However, NUQ works sufficiently good even without this regularization (see Table 6 in Section F.4). In all experiments on OOD detection, we use $\hat{U}_e(\mathbf{x})$ as a measure of uncertainty. An additional illustrative experiment on detecting actual aleatoric and epistemic uncertainties is presented in SM, Section E.

### 6.1 How NUQ Affects Model Predictions?

One may ask whether the nonparametric classification method used in NUQ, trained on some embedding from the base model, has any relation to the original neural network. To reassure the reader, we provide an argument that it well approximates the predictions of the base model and NUQ-based uncertainty estimates can be used for the base model as well. Specifically, we compute the agreement between predictions obtained from the Bayes classifier based on kernel estimate (i.e. the one used in NUQ) and base models' predictions. This metric formally can be defined as $\text{agreement}(\hat{p}, p) = \frac{1}{n} \sum_{i=1}^{n} I \left[ \arg\max_j \hat{p}(y = j \mid \mathbf{x}_i) = \arg\max_j p(y = j \mid \mathbf{x}_i) \right]$. For CIFAR-100 (see experiments with this dataset in Section 6.2.1), this metric gives us the agreement of 0.975, which shows that the approach is accurate. Additionally, we computed the aleatoric uncertainty for

| OOD dataset | MaxProb* | Entropy* | Dropout | Ensemble | TTA | Energy* | DUQ* | SNGP* | DDU* | NUQ* |
|---|---|---|---|---|---|---|---|---|---|---|
| SVHN | 79.7±1.3 | 81.1±1.6 | 77.6±2.5 | 82.9±0.9 | 81.6±1.2 | 62.0±1.7 | 88.7±6.3 | 86.2±7.4 | 89.6±1.6 | **89.7±1.6** |
| LSUN | 81.5±2.0 | 83.0±2.1 | 76.8±5.1 | 86.5±0.8 | 85.0±2.7 | 82.7±0.1 | 90.8±6.7 | 83.7±8.6 | 92.1±0.6 | **92.3±0.6** |
| Smooth | 76.6±3.5 | 77.8±5.2 | 63.3±3.8 | 83.7±1.2 | 73.2±10.8 | 71.5±4.6 | 91.1±8.4 | 60.9±12.5 | **97.1±3.1** | 96.8±3.8 |

Table 1: OOD detection for CIFAR-100 in-distribution dataset with the ResNet-50 neural network. The top two results are shown in bold and underline correspondingly. Evaluation is done for three models trained with different seeds to estimate the standard deviation. Methods requiring a single pass over the data to compute uncertainty estimates are marked with *.

| OOD dataset | MaxProb* | Entropy* | TTA | Energy* | Ensemble | DDU* | DUQ* | SNGP* | NUQ* |
|---|---|---|---|---|---|---|---|---|---|
| ImageNet-R | 80.4 | 83.6 | 85.8 | 78.34 | 84.4 | 80.1 | 73.3 | 85.0 | **99.5** |
| ImageNet-O | 28.2 | 29.1 | 30.5 | 60.0 | 51.9 | 74.1 | 71.4 | 75.8 | **82.4** |

Table 2: ROC-AUC score for ImageNet out-of-distribution detection tasks for different methods. Methods requiring a single pass over the data to compute uncertainty estimates are marked with *.

all test objects and found the average percentile of the objects with disagreement is $94.94 \pm 4.75$. Thus, disagreement appears for high uncertainty points as expected.

## 6.2 Image Classification

The main experiments with image classification are conducted on CIFAR-100 [36] and ImageNet [12]. However, several additional experiments with other image classification datasets such as MNIST and SVHN can be found in SM, Sections F.1 and F.2.

We compare NUQ with popular UE methods, which do not require significant modifications to model architectures and training procedures. More specifically, we consider Maximum probability (MaxProb) of softmax response of a NN, entropy of the predictive distribution, the Monte-Carlo (MC) dropout [21], an ensemble of models trained with different random seeds (deep ensemble), the Test-Time Augmentation (TTA; [48]), DDU [55], SNGP [46], DUQ [69], and an energy-based approach [47].

For Monte-Carlo dropout, ensembles, and TTA, we first compute average predicted class probabilities and then compute their entropy (see the ablation study in SM, Section F.8). More details can be found in SM, Section B. For deep ensembles, we fixed the number of models to 5. Additional experiments, where we changed the number of models, are presented in SM, Section F.5.

### 6.2.1 CIFAR-100

In this experiment, we test UE methods on the out-of-distribution detection task. We treat the OOD detection as binary classification (OOD/not-OOD) using only the uncertainty score. Following the setup from the recent works [69, 70, 65], we use SVHN, LSUN [73], and Smooth [28] as OOD datasets. The reported metric is ROC-AUC.

As a base model, we train ResNet-50 from scratch on the CIFAR-100 dataset. NUQ was applied to the features from the penultimate layer of the model, and the density estimate is given by GMM, as it provides the best results (see the results for other choices of hyperparameters in Section F.4).

The results are presented in Table 1. We can clearly see that NUQ and DDU show close results while outperforming the competitors with a significant margin.

### 6.2.2 ImageNet

To demonstrate the applicability of NUQ to large-scale data, we evaluate it in the OOD detection task on ImageNet [12]. As OOD data, we use the ImageNet-O [30] and ImageNet-R [29] datasets. ImageNet-O consists of images from classes not found in the standard ImageNet dataset. ImageNet-R contains different artistic renditions of ImageNet classes.

In contrast to the previous experiment, we found that for NUQ, it is more beneficial to use KDE as a density estimator $p(\mathbf{x})$, rather than GMM. Importantly, it took us approximately 5 minutes to receive uncertainties over all ImageNet datasets with a CPU, i.e. our NUQ implementation is readily applicable to large-scale data (see more details in SM, Section F.7).

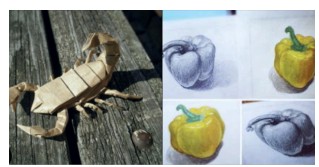 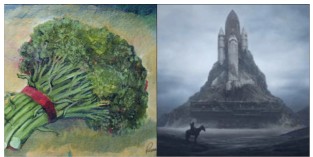 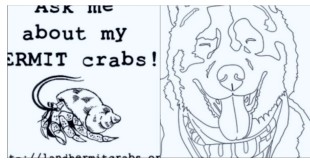

a) Low uncertainty  b) Medium uncertainty  c) High uncertainty

Figure 2: Typical OOD images from ImageNet-R ordered from low uncertainty (bottom 10%) (a) to high uncertainty (top 90%) (c) by the NUQ $\hat{\mathbf{U}}_e(\mathbf{x})$ scores. We can clearly see that low-uncertainty images resemble real-world objects presented in vanilla ImageNet.

The results are summarized in Table 2. We see that for ImageNet-O, many methods show good OOD detection quality, but NUQ achieves an almost perfect result. For ImageNet-R, simple approaches completely fail while DDU, SNGP, and DUQ perform well, and NUQ shows the best result with a large margin.

Note that unlike in the CIFAR-100 experiment, for ImageNet, NUQ significantly outperforms DDU. We conjecture that GMM struggles to approximate density here as an embedding structure is much more complicated for ImageNet compared to CIFAR-100 (see some visualizations in Section F.3). NUQ is beneficial in this case as KDE is much more flexible than GMM and provides a better result.

Additionally, we looked at some typical samples from ImageNet-R with low, moderate, and high levels of uncertainty as assigned by NUQ, see Figure 2. Here, low, medium, and high uncertainties correspond to 10, 50, and 90% quantiles of the epistemic uncertainty distribution for images from the ImageNet-R dataset. We observe that uncertainty values correspond well to intuitive degree of image complexity compared to the original ImageNet data. Some additional ImageNet experiments are presented in SM, Section F.9.

## 6.3 Text Classification

Experiments on textual data are performed in low-resource settings, where we train models on small subsamples of original datasets. This regime can be challenging for many UE methods, while NUQ is naturally adapted to it. We compare NUQ to the best performing methods on image classification: DDU and deep ensemble, and to the standard baselines: MC dropout and MaxProb.

The methods are evaluated on two tasks: OOD detection and classification with a reject option. Classification with rejection experiments are conducted on SST-2 [67], MRPC [17], and CoLA [71]. The evaluation metric is RCC-AUC [18]. Experiments with OOD detection are conducted on ROSTD [23] and CLINC [39], which originally contain instances marked as OOD, and a benchmark composed from SST-2 (in-domain), 20 News Groups [38], TREC-10 [44, 32], WMT-16 [5], Amazon [52] (sports and outdoors categories), MNLI [72], and RTE [11, 1, 26, 3], where SST-2 is used as an in-domain dataset, while the rest as OOD datasets. The final score is averaged across all OOD datasets. Evaluation metric is ROC-AUC as in the image classification task. The dataset statistics are presented in SM, Section G.

We use a pre-trained ELECTRA model with 110 million parameters. The features for NUQ and DDU are taken from the penultimate classification layer. The details of the model, hyperparameter optimization, and UE methods are presented in SM, Section G.

### 6.3.1 Classification with a Reject Option

Figure 3 presents the results for classification with rejection. For the MRPC dataset, we can note that NUQ, SNGP, and MC dropout show similar results to the MaxProb baseline. DDU stands out, demonstrating substantially worse performance in settings with small amount of training data.

A similar dynamics for DDU can be noted on the CoLA dataset, where it works substantially worse than the baseline when only 1% of the training data is available. SNGP in this experiment manages to reach other methods only when 10% of training data is unlocked. On contrary, NUQ is always on par or better than the baseline outperforming all other computationally efficient methods and has similar performance as computationally expensive MC dropout.

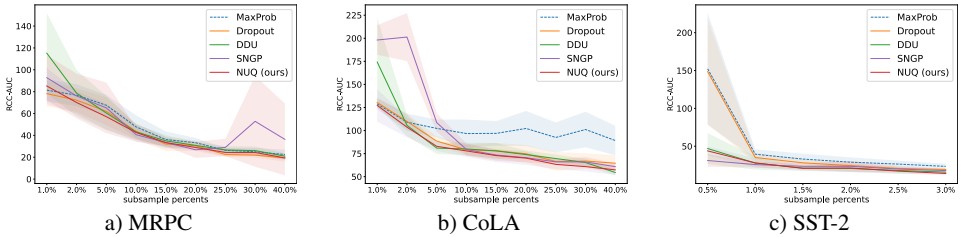

Figure 3: RCC-AUC↓ of classification with rejection depending on fraction of unlocked training data.

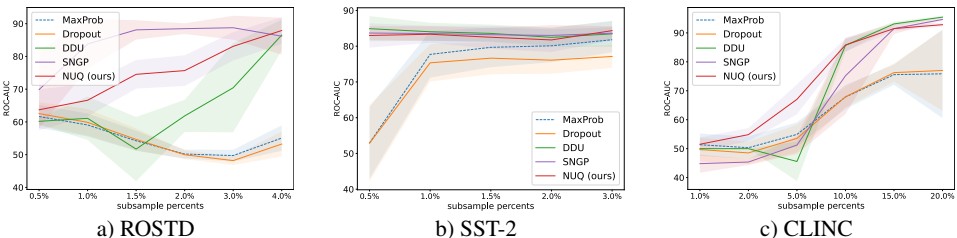

Figure 4: ROC-AUC↑ of OOD detection depending on the fraction of unlocked training data.

On SST-2, NUQ works similar to SNGP and DDU, substantially outperforming the MaxProb baseline, and MC dropout in the extremely low resource setting. Starting from 1.5%, all methods work similarly with small advantage over the baseline.

Overall, we can conclude that in the settings with a small amount of training data, NUQ can be the best choice for estimating uncertainty for the classification with a reject option: it works similar to other methods when there is much training data and does not deteriorate in the low-resource regime, demonstrating much better results than others. SNGP is able to approach NUQ on SST-2. However, its performance is not stable, which is illustrated by poor results on CoLA. DDU sometimes fails to outperform the baseline with small amount of training data, which might be due to its reliance on the assumption that training data has a Gaussian distribution, which does not hold in this setting.

### 6.3.2 Out of Distribution Detection

Figure 4 presents the results of OOD detection. We see that NUQ confidently outperforms MC dropout on all datasets and outperforms DDU on ROSTD and CLINC. This is especially notable for extremely low-resource settings. SNGP has some advantage over NUQ on ROSTD, but it substantially falls behind on CLINC. Moreover, unlike SNGP, NUQ always outperforms the MaxProb baseline, therefore, it might be a better choice for OOD detection in low-resource regimes. Finally, for ROSTD and CLINC datasets, DDU works poorly, sometimes failing to outperform the MaxProb baseline, which stems from the incorrect assumption about the Gaussian distribution of training data.

## 7 Conclusions

This work proposes NUQ, a new principled uncertainty estimation method that applies to a wide range of neural network models. It does not require retraining the model and acts as a postprocessing step working in the embedding space induced by the neural network. NUQ significantly outperforms the competing approaches with only the recently proposed DDU method [55] showing comparable results. Importantly, in the most practical example of OOD detection for ImageNet data, NUQ shows the best results with a significant margin. NUQ is also superior to DDU and other methods on text classification datasets in both OOD detection and classification with rejection. The code to reproduce the experiments is available online at `https://github.com/stat-ml/NUQ`.

We hope that our work opens a new perspective on uncertainty quantification methods for deterministic neural networks. We also believe that NUQ is suitable for in-depth theoretical investigation, which we defer to future work.

**Acknowledgements.** The research was supported by the Russian Science Foundation grant 20-71-10135

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
