# OpenReview forum: "Nonparametric Uncertainty Quantification for Single Deterministic Neural Network"
_NeurIPS.cc/2022/Conference — NeurIPS 2022 Accept_

### Official Review · Reviewer_RSJH · 2022-07-08

**Rating:** 6
**Confidence:** 5
**Soundness:** 3 good
**Presentation:** 2 fair
**Contribution:** 3 good

**Summary:**

This paper is about epistemic/aleatoric uncertainty quantification in single deterministic neural networks using density estimation in the feature space to measure epistemic uncertainty. In particular this work uses a non-parametric density estimation method (kernel density estimation) which makes less assumptions than other methods like DUQ and similar.

I believe the contributions are:
- The use of non-parametric kernel density estimation to model the density for epistemic uncertainty quantification in single deterministic neural networks.
- Experimental improvements in out of distribution detection on CIFAR100 and ImageNet benchmarks, and text classification with rejection.
- A good statistical theory argumentation on how the method is built, and proofs for the consistency of the proposed method in the case of classification with rejection.


**Questions:**

- In Line 127, how do choose the "properly chosen bandwidth" to make sure that the difference of eta's converges to the desired distribution? I think there has not been much discussion in the paper about how to select the bandwidth for the KDE part of this method.
- Have you evaluated other possible approximations for the kernel density estimate?
- Would you consider rewriting the paper to include binary/multi-class details in the paper, and leave derivations in the supplementary?


**Limitations:**

 As I mentioned in weaknesses, the paper does not mention that it uses an approximation for the kernel density estimate, and this could be a limitation that could be addressed in the future.

There are no other limitations mentioned in the paper about the proposed method, even as the checklist says the paper does discuss limitations (I do not see where, I might be wrong).

Overall I see no negative societal impact to discuss.


**Strengths And Weaknesses:**

Strengths
- The proposed method, first perform density estimation in feature space, and then uses statistical properties to derive estimates of aleatoric and epistemic uncertainty, overall these make sense, but I have not went into the statistical details in depth.
- There are good improvements in out of distribution performance as measured by ROC AUC both in CIFAR100 and ImageNet, and also in text classification and good comparisons for classification with rejection.
- I like that a non-parametric density estimation is used to estimate epistemic uncertainty in feature space, it is a simple change, previous method usually made implicit assumptions of Gaussianity or other shapes in the feature space, and it makes sense that performance improves when taking less assumptions. I think the authors implicitly make this argument in this paper.
- The paper seems to be correctly evaluated, there are OOD detection results on ImageNet (hard) and CIFAR100 (also hard), and for classification with rejection on text datasets on a variety of datasets.

Weaknesses
- I think the paper is hard to follow and read, in particular there are a lot of statistical details and some of the machine learning details can be lost in there. There is excessive use of alternative notation, for example $\eta$ is used for the conditional density p(Y = 1 | X = x), I think it is better to use probabilistic notation directly and not introduce additional greek letters as makes the notation hard to follow. Same can be said for risk definitions.
- Overall about readability, I think the paper should be rewritten to put the terms that are usable for implementation from a machine learning perspective (like binary and multi-class uncertainty components), and leave derivations and proofs for the supplementary. The audience of NeurIPS is largely machine learning people, and the paper notation and arguments could be simplified so it is easier to follow for this audience. The statistical details are important but after reading this paper, if I want to implement this method, to me it is not clear what parts I should use or directly how this method works conceptually.

~~- The results in the paper only present out of distribution detection results as main result, and classification with rejection on text datasets. There are additional results on the supplementary but I believe these are minor. Overall I think there are missed opportunities for evaluation, for example, calibration error of epistemic uncertainty (Figure 2 could be a start point).~~

~~- The authors argue that one advantage of this method is that it can disentangle aleatoric and epistemic uncertainty, but this is only vaguely evaluated (only in a toy example in the supplementary), and not in the main paper. I think this is a missed opportunity as this is an important topic that is often overlooked, and there are opportunities for comparison with Kendall and Gall. 2017, which is a well known baseline for uncertainty disentanglement. I recommend that the authors also consider how to evaluate the aleatoric and epistemic uncertainty separately and show that they behave as expected.~~

~~- Since density estimation is computationally expensive, the authors resort to an approximation of the kernel density estimate with nearest neighbors in feature space. I think this is fine, but there is no evaluation of approximation quality or ablations on how to select an approximation for nearest neighbors and kernel density estimates, the authors just use HNSW for this purpose.~~

Minor issues
- I think a diagram showing how the method works and its training/inference process would be an easy way to understand the overall proposed approach.

~~- I think the paper is missing references to the text datasets (CoLA, SST-2, and MRPC) and other datasets used in the text classification experiments.~~

~~- I am not sure what score is used for OOD detection, my guess is that it is the epistemic uncertainty $U_e$, and this should be explicit in the evaluation section.~~

---

> ### Author Response · Authors · 2022-08-02
> **Response to Reviewer RSJH**
>
> Dear colleague, we would like to thank you for insightful comments and suggestions on our paper. Below, we answer them one by one. Please note that in the updated version of the manuscript, in the main part we only fixed typos. All the additional clarifications were placed in SM, and we plan to incorporate some of them into the main part in the camera-ready version of the paper (if accepted) using the additional page provided on that stage.
>
> 1. **Comment:** The paper is hard to follow and read, in particular there are a lot of statistical details... Paper should be rewritten to put the terms that are usable for implementation from a machine learning perspective...
>
> **Answer:** We believe that statistical details are important for showing that the proposed approach is theoretically grounded. Most of the mathematical proofs are already moved to SM, while the main part contains the essential details. To improve readability and clarify some aspects related to implementation we are going to introduce the following changes to camera-ready (if accepted).
>
> a. We will add to the main part of the paper the scheme of the proposed method. The draft of this diagram can be found in Section G, SM.
>
> b. We will add the pseudo-code of the algorithm describing the actual computations performed on the inference stage to the main part of the paper. Currently, you can find it in Algorithm 2; see Section G in SM.
>
> c. We will also try to simplify notations as you suggested. You may already see our effort in this direction in Section G of SM.
>
> The additional page in the camera-ready version of the paper should be enough to incorporate this material to the main part of the paper.
>
> 2. **Comment:** I think there are missed opportunities for evaluation of calibration error of epistemic uncertainty (Figure 2 could be a start point).
>
> **Answer:** We have made plots by mixing vanilla ImageNet images with ImageNet-R/O. They are indeed illustrative, we added them to SM, Section K.
>
> 3. **Comment:** Disentangle of aleatoric and epistemic uncertainty is only vaguely evaluated…
>
> **Answer:** Thank you for this important question. Modern popular image datasets (like ImageNet or CIFAR10/100) *almost* do not contain an aleatoric uncertainty [1] because they are carefully assessed. Thus, there is no big room for us to demonstrate aleatoric uncertainty behavior apart from a toy example.
>
> Still, the main result of this paper with respect to uncertainty disentaglement is usefulness of epistemic uncertainty estimate by NUQ to detect OOD data compared to aleatoric uncertainty (MaxProb). Additionally, we  show that epistemic uncertainty also outperforms the aleatoric one on text data in low resource setting when aleatoric uncertainty estimate becomes unreliable. We will elaborate on this interpretation in the camera ready version based on your suggestions.
>
> Regarding the paper you mentioned, application of our approach to semantic segmentation and depth regression is a future work for us.
>
> [1] Pleiss, G. et al. Identifying mislabeled data using the area under the margin ranking. In NeurIPS 2020.
>
> 4. **Comment:** No evaluation of approximation quality or ablations on how to select an approximation for nearest neighbors and kernel density estimates
>
> **Answer** Thank you for the comment. We added the corresponding comments and experiments to SM, Sections H.2 and H.4. We see that there is a relatively wide range of suitable values, which yields good results.
>
> 5. **Comment:** The paper is missing references to the text datasets…
>
>     **Answer:** Thank you! We added references to all text datasets.
>
> 6. **Comment:** What score is used for OOD detection? My guess is that it is the epistemic uncertainty Ue...
>
>     **Answer:** Thank you for pointing this out. We will add an explicit note in the text to avoid misunderstanding.
>
> 7. **Comment:** Line 127, how do choose the "properly chosen bandwidth"...
>
>     **Answer:** The hyperparameter $h$ is tuned to maximize classification accuracy on training data for the Bayesian classifier based on Nadaray-Watson estimator of the conditional label distribution. Such a choice doesn't guarantee that the convergence condition of Eq 2 will be satisfied, however, it gives a good practical performance. We experimented with choosing a smaller bandwidth to ensure undersmoothing but it didn't give any boost in performance. We will add the remark on the choice of hyperparameter $h$, which we currently we put it in Section H.1 of SM.
>
> 8. **Comment:** Have you evaluated other possible approximations for the kernel density estimate?
>
>     **Answer:** We didn't try other approximations for KDE as HSWN already gave us very good performance in terms of quality and speed. However, we considered the choice of density estimate itself. Specifically, we studied different kernels (RBF and sigmoidal) as well as GMM (a Gaussian per class) approximation of marginal density. The results for CIFAR100 are presented in SM, Table 3.

---

> > ### Comment · Reviewer_RSJH · 2022-08-08
> > **Good reply but some small issues left**
> >
> > Dear Authors,
> >
> > Yes your reply does answer most of my points, I will update my review in the coming days. There are only a couple of points that I think deserve a bit more discussion:
> >
> > - **Calibration**: For this you would have to use standard metrics like expected calibration error and reliability plots, I am not sure how Figure 8 in the SM (Section K as pointed by the authors) relates to evaluation of probability calibration. Also I do not understand what Figures 4b/c and 8 in the SM mean, it is not clear what these plots are doing, further explanation is needed. This looks like a accuracy vs confidence plot (particularly 4a), but I cannot interpret the other plots.
> >
> > - **Aleatoric vs Epistemic Uncertainty**: It is true that datasets do not have aleatoric uncertainty information, but this is not needed, for OOD detection, you can evaluate aleatoric and epistemic uncertainty, aleatoric uncertainty should not be able to separate OOD from ID data, while epistemic uncertainty is able to. You can use this as a crude comparison of aleatoric vs epistemic uncertainty quality.

---

> > > ### Author Response · Authors · 2022-08-08
> > > **Additional comments to Reviewer RSJH**
> > >
> > > Dear reviewer RSJH,
> > >
> > > Thank you for your positive feedback!
> > >
> > > Below we answer your questions one by one.
> > >
> > > **1) Calibration**
> > > First of all, we apologize that our answer diverged significantly with your question due to the misunderstanding on our side. Nevertheless, we think that additional plots, we showed in Figure 8, are useful for better illustration of the proposed method’s performance (see the detailed explanation below).
> > >
> > > We note that in the setup of the Figure 2 in the main text, it is not straightforward to assess the quality numerically as the standard calibration measures are not directly applicable here due to two reasons:
> > > i) Many uncertainty scores (such as the NUQ-based estimate of epistemic uncertainty) are arbitrary real numbers, while to compute ECE or plot reliability diagrams, one needs probabilities, i.e., the numbers in [0,1].
> > > ii) The calibration is not fully defined for the out–of-distribution data as classes are often different from the ones for the training data.
> > >
> > > For the "soft" OOD data that are not very different from the training one, we can use the plots similar to Figure 4a in Supplementary Material to overcome the issue (i). However, for the OOD data that is significantly different from in-distribution data like ImageNet-R or -O, these plots are not meaningful due to (ii). Thus, due to the reasons above, we proposed such a visual proxy (Figure 2) to somehow evaluate the "epistemic uncertainty calibration".
> > >
> > > Coming back to Figures 4b/c and 8 in Supplementary Material, these plots should be understood as follows.
> > >
> > > **Preparation part** (on example of Figure 8): We pick two datasets (test set for in-distribution ImageNet and out-of-distribution Imagenet-[R, O]  for figures 8[a,b] correspondingly) and merge (concatenate) them. The resulting dataset has the size  len(test ImageNet) + len(ImageNet-[R, O]) and each object is labeled either 'i' - in-distribution or 'o' - out-of-distribution.
> > >
> > > Then we estimate some OOD scores (in the case of NUQ it is epistemic uncertainty) for each object of the concatenated dataset. Then we sort all these objects in ascending order according to these OOD scores.
> > >
> > > **Plotting part:** For each point on X-axis we consider the corresponding number of objects with lowest uncertainties from the dataset and plot on Y-axis the number of OOD points among those. Thus the ideal plot should have value zero until the point len(test ImageNet) (i.e., all the points with low uncertainty are from in-distribution data), and then it should behave like y = x - len(ImageNet). Thus, informally, the method which is closer to the ideal case (the lower curve) can be thought as the one showing better performance in OOD detection tasks. This plot complements ROC AUC scores and provides more intuition on the OOD detection method behavior.
> > >
> > > **2) Alearotic vs Epistemic**
> > >
> > > Thank you for a good idea about the disentanglement demonstration!
> > >
> > > We performed an additional experiment on ImageNet vs Imagenet-[R, O] using different types of uncertainty to detect an OOD for our kernel-based classifier.
> > > The results are as follows (ROC AUC scores):
> > > | Dataset    | Epistemic | Aleatoric |
> > > |------------|-----------|-----------|
> > > | Imagenet-O | 0.824     | 0.538     |
> > > | Imagenet-R | 0.995     | 0.843     |
> > >
> > > We indeed see that results for aleatoric uncertainty are way worse than the ones for epistemic uncertainty. We should admit that MaxProb (0.804 for ImageNet-R) and Entropy (0.836 for ImageNet-R) can also be considered as estimates of aleatoric uncertainty and show comparable results.

---

> ### Author Response · Authors · 2022-08-07
> **] Response to Reviewer RSJH II**
>
> Dear reviewer RSJH,
>
> Did you have a chance to take a look at our response, does it clarify your questions?

---

### Official Review · Reviewer_5BnR · 2022-07-11

**Rating:** 7
**Confidence:** 4
**Soundness:** 3 good
**Presentation:** 3 good
**Contribution:** 3 good

**Summary:**

This paper proposes Nonparametric Uncertainty Quantification (NUQ), a simple approach to estimate predictive uncertainty using kernel density estimators. The model uses kernel density estimators for prediction (Eq 1) and squared kernel magnitude (Eq 2) for predictive variance. The method adopts a simple decomposition between aleatoric and epistemic uncertainty.  A consistency proof for the method's ability in approximating the optimal learning-with-abstention risk is also provided. Experiments are conducted on large-scale vision benchmark (ImageNet) and several standard NLP tasks, where NUQ illustrated consistent improvement over other methods.

**Questions:**

Please see **Weakness** (1)-(3)

**Limitations:**

The method's technical clarity can use some improvement. Please see **Weakness**

**Strengths And Weaknesses:**

** Strength **
  * A simple and novel approach to deep uncertainty quantification grounded on kernel density theory.
  * The method comes with optimal guarantees towards minimizing certain learning-under-abstention loss.
  * The method was extensively evaluated on vision and language benchmarks, showing good performance.

** Weakness **
I only have some minor suggestions about technical clarity

 (1) Some of the technical details are missing or buried in the SM. For example, on line 129, what are $p(x)$ and $\mathbf{u}$? Please give brief explanation in the main text and then point to appropriate section / table in the SM.
 (2) In the experiments, how are $\sigma^2$ and $p(x)$ defined and estimated?
 (3) In the experiments, how are hyper-parameters such as $\sigma^2$ and $h$ estimated? Specifically, what objective are you minimizing? Also in practice, when the test data / OOD data are not available, how should practitioner set the hyper-parameters? The paper can benefit a section (maybe in SM) that collects all hyperparameters of the method and discuss practical strategies in setting them.

---

> ### Author Response · Authors · 2022-08-02
> **Response to Reviewer 5BnR**
>
> Dear colleague, we would like to thank you for your insightful comments and suggestions. Below, we answer them one by one. Please note that in the updated version of the manuscript, in the main part we only fixed typos. All the additional clarifications were placed in SM.
>
> 1. **Comment:** ``On line 129, what are $p(x)$ and $u$? Please give brief explanation in the main text and then point to appropriate section / table in the SM.''
>
>     **Answer:** $p(x)$ is a marginal distribution of the covariates that is a very important part of the obtained uncertainty estimate. It determines how much data we have in the vicinity of the particular data point and thus is expected to directly correlate with epistemic uncertainty. Our resulting formulas confirm the expected behavior. We have added the missing definition in the paper and will add the corresponding explanation to the camera-ready version of the paper (if accepted).
>
>     In Eq. 2, $u$ is just a free parameter over which the integration is done. This integral can be computed in the closed form. We provide the resulting values for some standard choices of kernels in Table 3; see Section C.1 in Supplementary Material.
>
> 2. **Comment:** ``In the experiments, how are $\sigma^2$ and $p(x)$ defined and estimated?''
>
>    **Answer:** For the binary case, $\sigma^2(x)$ is a variance of Bernoulli random variable: $\sigma^2(x) = \eta(x) (1 - \eta(x))$ for $\eta(x) = p(Y = 1 | x)$. For the multiclass case we consider the maximum of standard deviations for 1 vs all Bernoullis for all the classes; see the details in Section A in Supplementary Material.
>
>     Distribution $\eta(x) = p(Y = 1 | x)$ (and consequently $\sigma^2(x)$) can be estimated via Nadaraya-Watson estimator according to Eq 1. Marginal density of covariates $p(x)$  can be estimated via kernel density estimate. Alternatively, one can use the GMM density estimate. We have observed that the usage of GMM is beneficial for CIFAR-100, while for ImageNet, KDE works better. We present the ablation study on the choice of the estimate in Section E.4 in Supplementary Material, while the analysis of the differences between CIFAR-100 and ImageNet is given in Section E.3.
>
>     The full algorithmic description of the resulting inference procedure is given in Algorithm 2, see Section G in Supplementary Material.
>
> 3. **Comment:** ``In the experiments, how are hyper-parameters such as $\sigma^2$ and $h$ estimated? Specifically, what objective are you minimizing?''
>
>     **Answer:** The estimate for $\sigma^2(x)$ is discussed above. The hyperparameter $h$ is tuned to maximize classification accuracy for the Bayesian classifier based on the Nadaraya-Watson estimator of the conditional label distribution. The cross-validation over training dataset is used for this purpose. We will add the remark on the choice of hyperparameter $h$ to Section 2.3.2 in the camera-ready version of the paper, while currently we added the comment on it in Section H.1 in Supplementary Material.
>
> 4. **Comment:** ``Also in practice, when the test data / OOD data are not available, how should practitioner set the hyper-parameters?''
>
>     **Answer:** Our approach does not require test or OOD data to be available. As discussed above, the bandwidth $h$ can be chosen directly on the training dataset via the cross-validation procedure. It is a significant benefit of our method that greatly eases its application in practice.
>
>     In principle, the knowledge of explicit OOD data might help to improve the quality of detection of OOD samples similar to the ones used for choosing the parameters. However, if some other OOD points are present at test time, the OOD-based choice of parameters might be even harmful. Essentially, our current approach is based solely on training data, i.e., it uses minimal information and focuses on explaining the training data.
>
> 5. **Comment:** ``The paper can benefit a section (maybe in SM) that collects all hyperparameters of the method and discuss practical strategies in setting them.''
>
>     **Answer:** Thank you for this remark! It is a good idea, and we drafted the corresponding section in Supplementary Material (Section H).

---

> ### Author Response · Authors · 2022-08-07
> **Response to Reviewer 5BnR II**
>
> Dear reviewer 5BnR,
>
> Did you have a chance to take a look at our response, does it clarify your questions?

---

> > ### Comment · Reviewer_5BnR · 2022-08-08
> > **Clarification questions addressed.**
> >
> > Thanks authors fro the response. I believe my clarification questions are sufficiently addressed, and I don't have additional issues with the paper, and therefore maintain my original score.

---

### Official Review · Reviewer_TAXi · 2022-07-13

**Rating:** 6
**Confidence:** 3
**Soundness:** 3 good
**Presentation:** 2 fair
**Contribution:** 3 good

**Summary:**

In this paper, the authors propose a method for computing the aleatoric and epistemic uncertainty estimates using a kernel-based estimator for the conditional label distribution. The method is flexible as it can use any embeddings generated from a NN model. The authors also do empirical evaluation on two different tasks of OOD detection and classification with rejection to show the utility of their method.

**Questions:**

- Lines 24-26 "MaxProb represents only aleatoric uncertainty..." I think this is incorrect. MaxProb in my point of view captures the total uncertainty, including aleatoric and epistemic. I'd be happy to hear the counterpoint.

- Line 33 "Such methods are very perspective.." Seems like a typo

- Lines 42-43 "...some methods require computing covariance...". Please mention the approaches and cite them

- Eq 2, what's u exactly and how is this thing practically computed?

- Line 129-130  "standard deviation of data label" - what's this std dev exactly?

- There's additional work [1, 2] that looks at distilling uncertainty estimates as well as looks at classification with rejection using Bayesian decision theory framework. This should be discussed in the related work, and potentially the distillation method should be compared in the experimental section [2, 3].

- It'll be nice to see what the additional runtime looks like for generating the approximation to conditional label distribution and how does it scale wrt models/datasets.

References

[1] Vadera, Meet, Brian Jalaian, and Benjamin Marlin. "Generalized bayesian posterior expectation distillation for deep neural networks." UAI (2020).

[2] Vadera, Meet, Soumya Ghosh, Kenney Ng, and Benjamin M. Marlin. "Post-hoc loss-calibration for Bayesian neural networks." In UAI (2021).

[3] Malinin, Andrey, Bruno Mlodozeniec, and Mark Gales. "Ensemble distribution distillation." ICLR (2020).

**Limitations:**

I've highlighted the shortcomings in the previous sections.

**Strengths And Weaknesses:**

Strengths

- The overall method is very flexible in the sense that it can utilize any existing neural network model.

- The experiments provided by the authors cover a large set of baselines and models.


Weakness

- The paper is not very easy to follow, especially around methodology. I've added more comments on this in the next section.

- Although the authors have used deep ensembles, the authors haven't used the traditional model uncertainty as a metric for epistemic
uncertainty (also known as mutual information) in the comparison. The authors haven't used their distilled version either for comparison.

---

> ### Author Response · Authors · 2022-08-02
> **Detailed answer on your comments and suggestions**
>
> Dear colleague, we would like to thank you for your insightful comments and suggestions. Below, we answer them one by one. Please note that in the updated version of the manuscript, in the main part we only fixed typos. All the additional clarifications were placed in SM.
>
> 1. **Comment** For deep ensembles, authors haven't used the traditional model uncertainty as a metric for epistemic uncertainty..
>
> **Answer:** Thank you for this important remark. In fact, we used various reduction approaches for all ensemble-like methods: TTA, MC-dropout, and standard ensembles. Namely, we used mean maximum probability, the standard deviation of predicted probabilities, mutual information, and entropy. However, in the manuscript, we decided to focus only on one approach (entropy) to reduce clutter, since, in our experiments, it performed better on average. We now added the results for test time augmentation and ensembles on ImageNet; see Table 12 in SM, Section J. As we can see, entropy shows competitive results with mutual information.
>
> 2. **Comment:** Lines 24-26 "MaxProb represents only aleatoric uncertainty.." I think this is incorrect...
>
> **Answer:** Thanks for this comment, it is a very interesting point to discuss. The expected probability distribution indeed leads to the total uncertainty under the Bayesian paradigm. Importantly, the expectation is taken via the true posterior. However, in practice, only the approximation of the posterior is used. It leads to the additional uncertainty of epistemic nature due to this approximation. This type of uncertainty is usually ignored in the recent papers on uncertainty quantification for deep learning models.
>
> In the setup of this paper, we have only a single set of learned parameters, and this problem is even more clear. The trained neural network model often predicts high confidence outputs for the points far from the training ones (OOD points). That means that the model will have low MaxProb uncertainty (corresponding to aleatoric one as our derivation shows in the considered setup) at such points though it has no grounds for any predictions in that area (i.e., it should have high epistemic uncertainty). The epistemic uncertainty estimate derived in our work helps to account for uncertainty in these regions.
>
> 3. **Comment:** Line 33 "Such methods are very perspective.." Seems like a typo
>
> **Answer:** Thanks, we have corrected it.
>
> 4. **Comment:** Lines 42-43 "some methods require computing covariance". Please mention the approaches and cite them.
>
> **Answer:** We have added a reference to the work [1].
>
> [1] K. Lee et al. A simple unified framework for detecting out-of-distribution samples and adversarial attacks. In NeurIPS 2018.
>
> 5. **Comment:** Eq 2, what's u exactly and how is this thing practically computed?
>
> **Answer:** In Eq 2, $u$ is just a free parameter over which the integration is done. This integral can be computed in the closed form. We provide the resulting values for some standard choices of kernels in Table 3; see Section C.1 in SM.
>
> 6. **Comment:** Line 129-130 "standard deviation of data label" - what's this std dev exactly?
>
> **Answer:** For the binary case, it is just a standard deviation $\sigma(x)$ of a Bernoulli random variable: $\sigma^2(x) = \eta(x) (1 - \eta(x))$. For the multiclass case, we consider the maximum of standard deviations for one vs. all Bernoullies for all the classes; see the details in Section A in SM. All the formulas for computing uncertainty estimates are now summarized in Algorithm 2; see Section G in SM.
>
> 7. **Comment:** There's additional work [1,2] that looks at distilling uncertainty estimates...
>
> **Answer:** The research direction of ensemble distribution distillation is indeed interesting and promising. We will discuss the mentioned papers in the related work section.
>
> However, we believe the comparison with these methods is not necessary. We note that from a practical perspective, the main goal of distribution distillation approaches is to speed up computations while maintaining the quality provided by an ensemble. In the mentioned papers, distilled models usually work not better than the ensemble (with very few exceptions). However, some methods considered in our paper, including NUQ, outperform ensembles with a significant margin, so approximately the same gap with distilled models is expected. Also, unlike NUQ and its competitors considered in our paper, the distillation approaches require an ensemble to be available (i.e., additional overhead) and also require quite significant algorithmic efforts.
>
> 8. **Comment:** It'll be nice to see what the additional runtime looks like..
>
> **Answer:** We evaluated the training and inference time overhead. The results show that training overhead is less than 1\%, and inference overhead is less than 10\%. In contrast, for ensembles, the overhead would require computations that are several times longer than standard inference. We added the resulting Table 11 in SM, Section I.

---

> > ### Author Response · Authors · 2022-08-07
> > **Response to Reviewer TAXi II**
> >
> > Dear reviewer TAXi,
> >
> > Did you have a chance to take a look at our response, does it clarify your questions?

---

> > > ### Comment · Reviewer_TAXi · 2022-08-08
> > > **Reviewer's response**
> > >
> > > Thanks for engaging in the discussion. I am unable to see the revised manuscript on openreview, maybe we'll only be able to see the changes after decision notification.
> > >
> > > The authors have largely addressed all my questions, although I am still not clear on what u means in equation 2. Please make sure you add additional details around it in the final paper. Also, for the std deviation of labels, is this the empirical variance computed from train dataset?
> > >
> > > I'll be updating my score after the discussion period. Thanks, and good luck!

---

> > > > ### Author Response · Authors · 2022-08-08
> > > > **Additional comments to Reviewer TAXi**
> > > >
> > > > Dear reviewer TAXi,
> > > >
> > > > Thank you for your positive feedback!
> > > >
> > > > Regarding your remaining questions:
> > > > 1) The integral over $u$ is just a constant that depends only on the choice of the kernel function. In the updated version of the manuscript, we will put just constant C in equation (2) and then provide the formula for it and the detailed comments after the equation.
> > > >
> > > > 2) For the binary case, we estimate the variance by computing $\hat{\sigma}^2(x) = \hat{\eta}(x) (1 - \hat{\eta}(x))$, where we take $\hat{\eta}(x)$ to be equal to Nadaray-Watson estimator of the conditional label distribution. For the multiclass case, we consider the maximum of standard deviations for one vs. all Bernoullies for all the classes; see the details in Section A in Supplementary Material.

---

> ### Comment · Area_Chair_PeAE · 2022-08-08
> **Please respond to author feedback**
>
> Thank you for reviewing this paper. Could you respond to the author feedback, or at least acknowledge that you've read the reply? Does the author reply address your concerns?
>
> Best, AC

---

### Meta-Review · Area_Chair_PeAE · 2022-08-20

**Recommendation:** Accept
**Confidence:** Certain

**Metareview:**

Decision: Accept

This paper propose a method for computing the epistemic/aleatoric uncertainties using a kernel-based estimator. Empirical evaluations on OOD detection & "classification with rejection" demonstrate the benefits of the proposed approach.

Reviewers commended that the proposed method is flexible, easy to be adapted to any neural network architecture. However, in initial reviews, concerns were raised about (a) clarity, (b) uncertainty evaluations in experiments (e.g., calibration, epistemic vs aleatoric uncertainty).

In author feedback, authors provided more explanations and some additional experimental results, which addressed many of the reviewers' concerns. Still the question regarding calibration metrics is there.

After reviewer-AC discussions, it is concluded that we can accept this paper to recommend its contribution in OOD detection research domains. Still I'd suggest in revision the paper will benefit from an edit to improve clarity, and I'd encourage the authors to consider including calibration metrics. As a side note, for the "classification with rejection" task, I'd suggest the authors to include a baseline regarding selective classification, e.g., the method of https://arxiv.org/abs/1705.08500.

**Award:**

No

---

### Decision · Program_Chairs · 2022-09-14

Accept